# Educational Video Improves Knowledge about Outpatients’ Usage of Antibiotics in Two Public Hospitals in Indonesia

**DOI:** 10.3390/antibiotics10050606

**Published:** 2021-05-20

**Authors:** Fauna Herawati, Rika Yulia, Bustanul Arifin, Ikhwan Frasetyo, Herman J. Woerdenbag, Christina Avanti, Retnosari Andrajati

**Affiliations:** 1Department of Clinical and Community Pharmacy, Faculty of Pharmacy, Universitas Surabaya, Surabaya 60293, Indonesia; rika_y@staff.ubaya.ac.id (R.Y.); rifin.fin81@gmail.com (B.A.); ikhwanfrasetyo@gmail.com (I.F.); 2Department of Pharmacology and Clinical Pharmacy, Faculty of Pharmacy, Universitas Indonesia, Depok 16424, Indonesia; retnosaria@gmail.com; 3Laboratory for Developmental Psychology, Faculty of Psychology, Universitas Surabaya, Surabaya 60293, Indonesia; setiasih_siegit@yahoo.com; 4Department of Pharmaceutical Technology and Biopharmacy, University of Groningen, 9713 AV Groningen, The Netherlands; h.j.woerdenbag@rug.nl; 5Department of Pharmaceutics, Faculty of Pharmacy, Universitas Surabaya, Surabaya 60293, Indonesia; c_avanti@staff.ubaya.ac.id

**Keywords:** information media, video, patient’s knowledge, antibiotics use, antibiotic resistance

## Abstract

The inappropriate use or misuse of antibiotics, particularly by outpatients, increases antibiotic resistance. A lack of public knowledge about “Responsible use of antibiotics” and “How to obtain antibiotics” is a major cause of this. This study aimed to assess the effectiveness of an educational video about antibiotics and antibiotic use to increase outpatients’ knowledge shown in two public hospitals in East Java, Indonesia. A quasi-experimental research setting was used with a one-group pre-test—post-test design, carried out from November 2018 to January 2019. The study population consisted of outpatients to whom antibiotics were prescribed. Participants were selected using a purposive sampling technique; 98 outpatients at MZ General Hospital in the S regency and 96 at SG General Hospital in the L regency were included. A questionnaire was used to measure the respondents’ knowledge, and consisted of five domains, i.e., the definition of infections and antibiotics, obtaining the antibiotics, directions for use, storage instructions, and antibiotic resistance. The knowledge test score was the total score of the Guttman scale (a dichotomous “yes” or “no” answer). To determine the significance of the difference in knowledge before and after providing the educational video and in the knowledge score between hospitals, the (paired) Student’s *t*-test was applied. The educational videos significantly improved outpatients’ knowledge, which increased by 41% in MZ General Hospital, and by 42% in SG General Hospital. It was concluded that an educational video provides a useful method to improve the knowledge of the outpatients regarding antibiotics.

## 1. Introduction

WHO reported that microbial resistance to antibiotics is a global health problem [1]. It is stated in their *WHO report on surveillance of antibiotic consumption* that the incidence of antibiotic resistance has increased rapidly in Asia, with the highest incidence in Southeast Asia [2,3]. Overprescribing [4], and the use of monotherapy broad-spectrum antibiotics [5,6] are the main causes of antibiotic resistance [7,8]. The reasons for antibiotic overprescribing and the use of monotherapy broad-spectrum antibiotics are the massive antibiotics revenue [9,10], the lack of supervision in antibiotic distribution by the regulatory authorities [11], physician-related factors, and patient-related factors [12]. Basic health research (Riset Kesehatan Dasar) from the Ministry of Health of the Republic of Indonesia [13] reported that 35.2% of Indonesians keep drugs on hand for self-medication, not only antibiotics but all medicines, and that 86.1% of the population store antibiotics that were obtained without a prescription [14,15].

Various measures need to be taken to prevent the development of antibiotic resistance and to reduce the spread of it [16]. One important aspect is to educate the public on how to use antibiotics correctly, and to convince them not to buy antibiotics without a prescription [17]. The most important patient-related factor is the lack of knowledge about antibiotics and their responsible use [18]. The Indonesian people perceive that antibiotics are “super drugs” for any possible disease, and that they can cure minor ailments caused by viruses, such as flu, colds, and fever [18,19].

Therefore, it is necessary to increase the patients’ knowledge by providing reliable information and education about antibiotics and their use [17,20]. In addition, it is not only important to consider the required content of the information that should be delivered to the patients, but also the information media used for conveying the message [21]. Various types of information media exist, including visual, auditory, and visual–auditory media [22]. A scoping review methodology reported positive results from video-based educational interventions (animated presentations, professionals in practice, and patient narratives); the effect differences between printed material and verbal education were statistically significant. Animated video formats offered advantages because elements are relatively easy to add or remove, content can be modified easily and they are flexible enough to accommodate clinical practices. Patients also tend to be more receptive to animated videos. In several studies, animated videos consistently showed improvement in short-term outcomes such as the knowledge and comprehension of the information provided by the healthcare team [23]. A video used to teach patients/parents about the appropriate use of antibiotics was shown to be more effective than a pamphlet. It increased knowledge and improved behavior [24,25]. It also showed longer-term knowledge retention, the post-intervention survey scores remaining high [26,27]. Recent studies carried out in Indonesia showed that watching an informative video positively influenced a patient’s or caregiver’s knowledge and perception of antibiotic use [18,28].

### 1.1. Aim of the Study

This study aimed to assess the effect of providing information in the form of an animated video containing information about antibiotics, to improve outpatients’ knowledge.

### 1.2. The Impact on Practice

Antibiotics should only be taken when prescribed by a physician. However, there is no strict regulation preventing patients from buying antibiotics without a prescription in pharmacies in Indonesia. The assumed benefits of antibiotics may cause a patient to purchase them for every symptom, even for minor ailments. The patients’ understanding of how to use antibiotics responsibly and how to obtain antibiotics correctly is still very poor in Indonesia. Self-medication behavior in this respect is dangerous and may pose a serious threat to the development of microbial resistance to antibiotics. This study shows that an educational video increases the patients’ knowledge about the responsible use of antibiotics, and about procedures for obtaining antibiotics correctly. This increase in knowledge may help to reduce the risk of antibiotic resistance development.

## 2. Results

The study on the effect of the educational video was carried out to improve patients’ knowledge on the responsible use of antibiotics and procedures for obtaining antibiotics. The characteristics of the respondents from MZ General Hospital and SG General Hospital are provided in Table 1. The number of females visiting the two hospitals was higher than that of males (52% in MZ General Hospital and 74% in SG General Hospital). The age distribution of the outpatients in the range of 18–45 years was 97% for MZ General Hospital, and 59% for SG General Hospital.

Table 2 shows that the total score of patients’ knowledge about antibiotics before the intervention (pre-test) was 58.8 in MZ General Hospital and 62.6 in SG General Hospital, while after the intervention the score was 82.7 at MZ General Hospital and 88.8 at SG General Hospital. Knowledge improvement was significant (*p* < 0.05) in both hospitals.

Among the five domains, there were two showing a significant improvement after intervention: “Obtaining the Antibiotics” (MZ: from 44.9 to 77.6; SG: from 53.1 to 95.8; *p* < 0.05) and the “Directions for Use” (MZ: from 64.3 to 81.6; SG: from 63.9 to 91.5; *p* < 0.05) (Figure 1).

The results of the pre-test of question Q1 showed that respondents at MZ General Hospital already had adequate knowledge that antibiotics are used for bacterial infections, with a score of 95.9. The respondents also had a good knowledge that antibiotics should be taken regularly (the pre-test score of Q10 was 73.5). On the other hand, the respondents at the SG General Hospital already knew that antibiotics would be ineffective when stored in a place exposed to sunlight, and also understood that being infected with antibiotic-resistant bacteria may result in higher costs of treatment (pre-test scores of Q14 and Q18 were 70.8).

Video intervention significantly improved the patient’s knowledge that antibiotics are medicines that can be purchased without a doctor’s prescription. This is obvious from the increased score of 40.8 for question Q6 at MZ General Hospital. At SG General Hospital, a significant improvement was seen in the knowledge about the fact that left-over antibiotics (e.g., from relatives) cannot be used in the case that the patient has similar symptoms (increased score of 42.7 for question Q7).

The respondents from both hospitals misunderstood the indicators for antibiotics. This incorrect knowledge is related to question Q1 and Q2, i.e., antibiotics are medicines used not only for diseases caused by bacterial infections, but also for diseases with symptoms of fever, runny nose, and sore throat. The knowledge score before the intervention was 95.9 (Q1) and 29.6 (Q2) at MZ General Hospital; 50.0 (Q1) and 56.3 (Q2) at SG General Hospital.

The paired Student’s *t*-test comparing pre- and post-intervention in each hospital revealed that the difference in the knowledge score before and after the intervention was statistically significant (*p* < 0.05) (Table 2).

## 3. Discussion

Our study to reduce the inappropriate use of antibiotics in outpatients was performed by providing information in the form of an educational video prior to dispensing the medication to outpatients at two General Hospitals in two regencies in the East Java Province of Indonesia, followed by evaluating the effect of watching the video. The results of the current study are in line with those from previously reported studies on a similar topic. Earlier studies showed that a “personalized” video intervention about responsible antibiotics use (provided in the local language) is useful to enhance public awareness on this topic [27,28,29]. This was especially shown for the use of antibiotics (or not) to treat specific diseases, and for parents taking care of their children and being too eager to give them antibiotics [27].

The outpatient age distribution in this study reflects an “age–sex pyramid” population. The growth rate in the S-regency population (1.10) was higher than in the L-regency population (0.02). The distribution of various age groups in the S-regency population forms the expansive shape of a pyramid, showing that the population is growing, while an equal proportion in each age group of the L-regency population points to a stationary population pyramid. The difference between the female and male ratios may be ascribed to the differences in regional population and the characteristics of the two hospitals. Although both are general hospitals, the focus of the two hospitals is slightly different. SG General Hospital has a beauty clinic (aesthetics) and serves a larger female population, comparable with the larger female population in the region.

The knowledge increment about antibiotics of the respondents after watching the video in the S regency (23.9) was lower than that of respondents in the L regency (26.2). The results of the post-test of respondents at the L regency were significantly higher than those of the post-test of respondents at the S regency (*p* = 0.001). Similar to Schoen’s study outcome [30], but different from Hjorth-Johansen’s [31], our study shows that a higher baseline knowledge produced a higher knowledge increment.

In Indonesia, antibiotics are a prescription-only medicine. They bear a circle with a red color logo on the packaging, but patients with a lack of knowledge are unaware of this and purchase antibiotics for every symptom, even for minor ailments, thereby assuming the general benefits of antibiotics. In both hospitals, the initial outpatients’ knowledge about how to obtain antibiotics was low (45.2 and 53.0). The outpatients did not know that antibiotics are prescription-only medicines (POM) [32] and are only indicated for the treatment of infectious diseases caused by bacteria. This situation is a yet-unknown threat from the community because people use antibiotics heedlessly, thereby unintentionally bringing harm to themselves and to others. The respondents’ knowledge about obtaining antibiotics is associated with their lack of knowledge about antibiotic resistance [33].

Research showed that health education regarding the use of an assistive device is more effective than lectures because the animation video increases people’s engagement and interest [34,35,36]. An educational video may improve the patients’ knowledge to a greater extent, so that people understand the content and can correctly answer when they are subsequently tested. Good knowledge provides better understanding, since regular exposure to accurate information can raise awareness and is likely to change behavior. Accordingly, educational video materials can yield many benefits as a tool for health prevention and promotion programs in the community, particularly concerning infectious diseases [26].

The patients’ baseline knowledge about correct indications for antibiotics and antibiotic resistance varied even in the same typical area, as the difference of the respondent pre-test score in both hospitals was statistically significant. After watching the five-minute education video, a patient’s knowledge may increase substantially. Besides this, many patients do not know that antibiotics are prescription-only medicines, but that there are no strict regulations for buying antibiotics without a prescription in Indonesia [37]. An immediate result of education is the increment of knowledge before and after the delivery of the information. This study cannot evaluate any patient behavioral change because a behavioral change is at level 3 of the benefit of education. The Kirkpatrick Model is a globally recognized method of evaluating the results of training and learning programs. It assesses both formal and informal training methods and rates them against four levels of criteria: reaction, learning, behavior, and results [38]. To become a habit, we recommend that healthcare practitioners provide this type of education regularly or each time a patient receives antibiotics. Customization is a long process, and a successful strategy for the rational use of medicines in any district in Java needs a combination of activities, i.e., educational, managerial, and regulational change [39,40].

### Limitations

There are several limitations to our study. First, it was carried out in a regional setting. The study results are limited to two hospitals with their own specific characteristics in East Java, and do not represent the general Indonesian population. In total, there are thirty-nine government-sponsored general hospitals in East Java that cover thirty-eight regencies/cities. However, the results of our study may be seen as good practice for other hospitals. Second, it shows a seasonal result because of a purposive sampling strategy during the study period [41]. Third, there may be an influence of respondent self-reports because of face-to-face education (Hawthorne effect) [42]. Fourth, there is uncertainty as to longer-term changes in patient behavior, because of a short exposure to the education material [24].

## 4. Materials & Methods

### 4.1. Study Design

A one-group pre-test—post-test design in quasi-experimental research was conducted to determine the effect of the intervention on the participants. The participants in this study were recruited with a purposive sampling technique. The inclusion criterion was an outpatient receiving an antibiotic prescription. All participants who met this inclusion criterion and gave verbal consent were included in the study.

All participants who gave consent completed a pre-test to assess their initial condition (Appendix A). All participants were requested to answer a similar questionnaire before (pre-) and after (post-) watching the educational video from a laptop. The participants watched the video in the waiting room, one participant for every session, privately. The length of the video was 4 min and 40 s. It started with an opening section, introduction, information about infections, antibiotics definition, the procedure to obtain antibiotics, antibiotic administration, antibiotic storage, antibiotic resistance definition and prevention, infection transmission prevention, and ended with a compelling visual that ties to a take-home message at a closing section (Appendix B). The video can be watched via the link: https://youtu.be/UFa3YS5xhAQ, accessed on 18 May 2021 [43].

The study population consisted of outpatients in MZ General Hospital and SG General Hospital that met the inclusion criteria of visiting the hospital for infectious disease, and receiving an antibiotic prescription. The study was done within the period of November 2018 to January 2019. Both the MZ General Hospital and SG General Hospital are the largest hospitals in the S regency and L regency, located 22 km and 43 km, respectively, from Surabaya, the capital city of East Java Province. In Indonesia, there are primary care centers and referrals to secondary or tertiary care facilities. General practitioners work at the primary care center, whereas specialist doctors work at secondary or tertiary care centers. This study was held in the outpatient clinic of two secondary care centers.

The questionnaire was developed based on existing literature [40] and consisted of 19 questions categorized in five domains: five items under “Definition of Infections and Antibiotics”, three items under “Obtaining the Antibiotics”, four items under “Directions for Use”, three items under “Storage Instructions”, and four items under “Antibiotic Resistance”. The questionnaire included all questions used, but not every question was applicable for both hospitals. When assessing face validity, several questions were dropped based on a good Cronbach alpha (reliability test) outcome. The reliability of the research questionnaire is considered to be good; the Cronbach alpha value was more than 0.6 for 10 item questions in MZ General Hospital, and 14 item questions in SG General Hospital [44]. In this study, the Cronbach’s alpha was 0.742 (MZ General Hospital) and 0.762 (SG General Hospital). The face-to-face interview data collection method was used before and after watching the video.

The following formula was used for calculating the adequate sample size in the prevalence study, wherein n is the sample size, *Z* is the standard normal variate at 5% type I error, *p* < 0.05 (1.96), *P* is the expected prevalence (0.5), and d is the precision of the effect size (0.1) [45]. There were 98 outpatients at MZ General Hospital, and 96 outpatients at SG General Hospital, who participated in the study.
(1)$$n=\frac{Z^{2}P\;\left(1-P\right)}{d^{2}}$$

### 4.2. Statistical Analysis

The total knowledge test score was a cumulative score on the Guttman scale, where the respondents selected a “yes” or “no” answer for each individual item question. The answers were analyzed descriptively. The percentage of the correct answers is reported in the tables below. To determine the significance of the difference in knowledge before and after providing the educational video, a paired Student’s *t*-test was applied. To examine the difference in the knowledge score between hospitals, Student’s *t*-test was applied.

## 5. Conclusions

The use of educational videos may increase a patient’s knowledge and awareness about the appropriate use of antibiotics. An educational video will improve patients’ short-term knowledge about the purchase and correct use of antibiotics, in order to reduce microbial resistance. However, one-time education will not definitely change behavior. Education must be provided continuously until it becomes habitual to use antibiotics correctly and responsibly. Besides this, stricter regulation is needed to avoid dispensing antibiotics without a prescription. A successful strategy for advancing the rational use of antibiotics is a combination of educational, managerial, and regulatory measures.

## Figures and Tables

**Figure 1 antibiotics-10-00606-f001:**
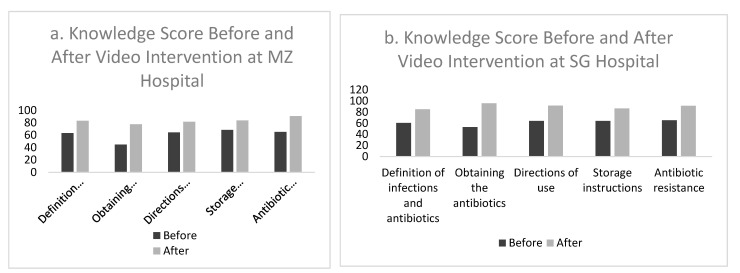
(**a**) Score of patient’s knowledge (%) before and after video intervention at (**a**) MZ General Hospital, and (**b**), SG General Hospital.

**Table 1 antibiotics-10-00606-t001:** Respondents’ demographic characteristics.

Characteristic	MZ General Hospital (*n* = 98)	SG General Hospital (*n* = 96)
	*n* (%)	*n* (%)
Gender		
Male	47 (48.0)	25 (26.0)
Female	51 (52.0)	71 (74.0)
Age (years old)		
18–25	12 (12.2)	8 (8.3)
26–35	36 (36.7)	12 (12.5)
36–45	47 (48.0)	37 (38.5)
46–60	3 (3.1)	38 (39.6)
>60	0 (0)	1 (1.0)

**Table 2 antibiotics-10-00606-t002:** Knowledge score test difference between hospitals.

Questions		MZ General Hospital			SG General Hospital			Δ *p*-Value
		Pre	Post	Δ	Pre	Post	Δ	
Domain: Definition of Infections and Antibiotics		63.3	83.2	19.9	60.4	84.9	24.5	0.278
Q1	Antibiotics are medicines used to treat diseases caused by bacterial infections.	95.9	98.0	2.0	50.0	90.6	40.6	0.000
Q2	Antibiotics are remedies for diseases with symptoms of fever, runny nose, and sore throat.	29.6	68.4	38.8	56.3	78.1	21.9	0.049
Q3	Amoxicillin/ampicillin/ciprofloxacin/cefixime/chloramphenicol/rifampicin/tetracycline/erythromycin are antibiotics.	N/A	N/A	N/A	68.8	88.5	19.8	-
Q4	Constant use of hand sanitizer or soap before doing an activity can prevent infection transmission.	N/A	N/A	N/A	N/A	N/A	N/A	N/A
Q5	Wearing a face mask when suffering a cough, cold or flu, will prevent infection transmission.	N/A	N/A	N/A	66.7	82.3	15.6	-
Domain: Obtaining the Antibiotics		44.9	77.6	32.4	53.1	95.8	42.7	0.125
Q6	Antibiotics are medicines that can be purchased without a doctor’s prescription.	40.8	81.6	40.8	N/A	N/A	N/A	-
Q7	If the disease has the same symptoms as a relative or a friend has, the patient can use the antibiotics left over by the relative or friend.	59.2	78.6	19.4	53.1	95.8	42.7	0.004
Q8	Antibiotics can be purchased from supermarkets or drug stores.	N/A	N/A	N/A	N/A	N/A	N/A	N/A
Domain: Directions for Use		64.3	81.6	17.3	63.9	91.5	27.2	0.042
Q9	If the condition has improved, the amount or dose of antibiotics to be taken must remain the same until the entire course of antibiotics is complete.	35.7	73.5	37.8	67.7	100	32.3	0.474
Q10	Antibiotics must be taken every day following the schedule directed by the doctor or the pharmacist until the course of antibiotics is finished.	73.5	91.8	18.4	60.4	83.3	22.9	0.577
Q11	Failure to comply with the antibiotics’ directions used as suggested by the doctor or pharmacist leads to an incomplete or no recovery from the disease.	N/A	N/A	N/A	64.6	91.7	27.1	-
Q12	Consumption of a food or beverage that the doctor or pharmacist recommends avoiding during the antibiotic course can reduce the efficacy of the drugs.	N/A	N/A	N/A	N/A	N/A	N/A	N/A
Domain: Storage Instructions		68.4	83.7	15.3	63.9	86.6	22.3	0.288
Q13	The remaining antibiotic tablet or syrup can be stored and used again if the same disease occurs.	55.1	71.4	16.3	58.3	90.6	32.3	0.056
Q14	Antibiotics will be ineffective when stored in a place exposed to sunlight.	68.4	83.7	15.3	70.8	88.5	17.7	0.770
Q15	Antibiotics can be stored in a freezer.	N/A	N/A	N/A	63.54	81.25	17.71	-
Domain: Antibiotic Resistance		65.3	90.8	25.5	65.2	91.2	25.5	0.998
Q16	Stopping the use of antibiotics before completing the course of treatment recommended by the doctor can cause the bacteria to become resistant.	69.4	86.7	17.4	59.4	86.5	27.1	0.245
Q17	When bacteria become resistant to antibiotics, the duration of the antibiotic course will not be affected.	54.1	93.9	39.8	N/A	N/A	N/A	-
Q18	Being infected with antibiotic-resistant bacteria can result in higher costs of treatment.	N/A	N/A	N/A	70.8	93.8	22.9	-
Q19	Being infected with antibiotic-resistant bacteria can have a deadly outcome.	N/A	N/A	N/A	66.7	92.7	26.0	-
Total		58.8	82.7	23.9	62.6	88.8	26.2	-

∆ *p*-values refer to Student’s *t*-test of ∆ values; ∆ values represent the differences in knowledge scores of hospital respondents before and after intervention. N/A: not available.

## Data Availability

No new data were created or analyzed in this study. Data sharing is not applicable to this article.

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
