# Peer review of "Educational Video Improves Knowledge about Outpatients’ Usage of Antibiotics in Two Public Hospitals in Indonesia"

_antibiotics, 2021, doi:10.3390/antibiotics10050606_

Round 1

Reviewer 1 Report

I think this is an interesting study about the chance that education of hospital outpatients can represent for antibiotics’ correct usage.

Still, I would make some significant adjustments:

  • Lines 73-76: I think this paragraph about what is a video as a media is redundant for the audience
  • Lines 82-95: these are all significant examples about the efficacy of video formats, but I would shorten this part
  • Lines 118-120: what were the inclusion criteria? These should be stated clearly in the text
  • Lines 130-132: Cronbach alpha values should probably be presented in the Result section
  • Lines 152-154: Age and gender distribution were very different in the two hospitals. Were there any differences in the services or the context of the two hospitals? If so, there should be probably more information about these characteristics in the text.
  • Table 2:
    • I think a caption for this table is necessary. ∆p-values seem referred to the difference between hospitals, while the ∆ values are merely the differences within hospitals.
    • The results of Student’s t-test between before and after scores, partially cited in the following text, should also be provided in the table.
    • The answers for many questions are not available in both hospitals. There should be an explanation for this in the text.
  • Lines 193-202: This paragraph provides almost the same information of the Study Design section, and should not be in the Discussion section
  • Lines 229-231: what exactly is meant here for “healthcare”? Who should be responsible in the Authors’ opinion to provide this education?
  • Lines 241-244: this paragraph is not part of the conclusions to be drawn from the study. I think this should be removed.
  • Appendix 1: I think that it should be stated that these questions are the same provided in English in table 2
  • Appendix 2: the video transcript should be provided also in English.

Reviewer 2 Report

Thank you for the opportunity to review this article.

This manuscript presents an educational intervention to improve patient’s knowledge about antibiotic use.  This is a relevant topic. However this manuscript lacks an overview about the current knowledge about educational videos for improving antibiotic knowledge and new insights.  Methods section lacks info to understand choices made in the setup. Discussion lacks references and comparing to relevant similar studies. Limitation section lacks critical reflection about the study.

Some major revisions are needed.

The manuscript needs English language revisions.

Please see my other comments below:

Abstract:

You state: “ Irrational use or misuse of antibiotics, particularly by outpatients”

For me irrational is not the correct term to refer to the patients’ use of antibiotics. Irrational means: not using reason or clear thinking, which is not completely correct if you talk about patients not using their medication correctly, because they lack the knowledge or the context is difficult for example.

In the abstract you use the term: purposive sampling technique. In the study design section it’s not mentioned. It’s not clear how you approached this sampling technique and why. And it is not mentioned with the limitations.

Line 36: Many patients are unaware that antibiotics are medicines that are only prescribed by physicians.

This is contradictory with stating that  patients can buy them without a prescription. Do you mean SHOULD only be taken  when prescribed by  a physician?

I suggest to first state there is free access to antibiotics in Indonesia. And then the reason why patients do it anyway: the assumed benefits of antibiotics may cause a patient to purchase them for every  symptom, even for minor ailments.

Line 51: Inappropriate is a better term than irrational prescribing. Physicians have all sorts of reasons (not always the correct ones) why they do prescribe. Physicians are not necessarily irrational.

Line 53: what do you mean with: weak distribution supervision?

Line 51: You bring together reasons for antibiotic resistance that have a link but are not necessarily the direct reason for antibiotic resistance. The overprescribing and broad spectrum use are the reasons for resistance. But the massive antibiotic sales, the patient- AND physician related factors (which you did not mention) are the reason for the overprescribing and use of broad spectrum. But you can’t place them at the same level in the same sentence.  Also I don’t think that “society” is the reason for antibiotic resistance. Societal issues and cultural habits are.

Line 54: “The latter are the most common causes of antibiotic resistance in the community.” Do you refer to the patient related factors? And do you mean than ‘the taking of antibiotics without a prescription’? It’s not clear for me what you mean. Which patient related factors?

Line 57: 35.2% of the Indonesians keep drugs for self-medication. Do you mean antibiotics? Or all sorts of medication? Because why is there a larger number that stores antibiotics? This is also self medication, no?

Line 66: a ‘super drugs’: a super drug or super drugs without “a”.

Line 69: I’m not a native English speaker myself. But I think: “It is thus necessary” should be changed in : “therefore it’s necessary”

Line 70: Edit the English in this phrase:  “It is also important to not only to consider the required content of the information to be delivered”

Line 72: I would delete these sentences: “Various types of information media exist, including visual, auditory, and visual-auditory media. Video as a digital media shows an arrangement of images that are seen and read sequentially at certain times to  provide illusions, images, and fantasies associated with the moving images. The eyes and ears will capture these to synthesize the information in the brain. “ It’s not relevant and common sense.

Line 87-94: I would delete this alinea. For me this is not relevant information for your research. I think there are other studies who used videos for educating patients about antibiotic use. And would rather know these results. What do we know what works already, what doesn’t. What knowledge do we lack?

Line 109. It think it’s better to use participants instead of  “research subjects”.

Line 110: Written consent? Verbal consent?

Line 119: the inclusion criteria, which ones?

Does this mean they watched it after their consultation? Did the physicians possibly educate the patients also during the consult?

Line 120: Is the hospital for infectious disease, a primary care center? Is there something like general practice in Indonesia? Or are all patients seen in hospitals?

Line 130: How was  the research questionnaire developed? Based on literature? Validated?

Results

Did you collect any other respondents demographic such as educational background? And is there a relationship with your results?

Table 2: Why are some questions not available in both hospitals?

Line 173: The intervention significantly improved the patient’s knowledge that antibiotics are medicines that can be purchaseD without a doctor’s prescription. Please comment on this in the discussion session. This is a rather disturbing result of your study!

Line 189

The goal of the study was not to reduce irrational use of antibiotics in outpatients. Because you studied the knowledge, you did not study if there was any behavioral change afterwards.

Line 193-201: not relevant information for your discussion session. I would rather know why you have chosen 2 similar regions? And not two completely different regions, to be able to compare. Because you have compared them.

Limitations:

Please elaborate on your limitations. For example you tested knowledge but not behavior change. You tested short term memory of participants but not long term knowledge. You only had participants who received already a prescription, this could influence your results. You lack a process evaluation to see how patients experienced this intervention.

Please provide English translations of your patient questionnaire and material.

Round 2

Reviewer 1 Report

The authors significantly applied almost all of the suggested changes to the original manuscript and I think it has considerably improved.

However I think the content of response 8:

  • (“The questionnaire contained all questions used, but not each question was applicable for both hospitals. Hence, some questions were not given for one of the hospitals because the question drops from the original set. When doing face validity, several questions were dropped based on a good Cronbach alfa (reliability test).”

should also be put in the published manuscript.

Reviewer 2 Report

It still lacks an overview about the current knowledge about educational videos for improving antibiotic knowledge in the introduction section. Or comparisons to similar studies in the discussion section. Aren't there other studies who used videos for educating patients about antibiotic use?  What do we know what works already, what doesn’t, but specifically  on the topic of antibiotic prescribing. What knowledge do we lack?

The limitations section misses crital reflection on the research and is too limited.

Answers provided in the review report are not always included in the paper.

The English language needs editing.

Round 3

Reviewer 2 Report

Thank you for your work ad extensive rewriting of your manuscript.